# Norcantharidin Induces Immunogenic Cell Death of Bladder Cancer Cells through Promoting Autophagy in Acidic Culture

**DOI:** 10.3390/ijms23073944

**Published:** 2022-04-01

**Authors:** Lili Xu, Bijia Su, Lijun Mo, Chenye Zhao, Zhenlin Zhao, Hongwei Li, Zhiming Hu, Jinlong Li

**Affiliations:** 1Institute of Biotherapy, School of Laboratory Medicine and Biotechnology, Southern Medical University, Guangzhou 510515, China; xulili21810226@sina.com (L.X.); subijia@sohu.com (B.S.); molijun304@sina.com (L.M.); hongwei3@yahoo.com (H.L.); 2Shenzhen Ruipuxun Academy for Stem Cell & Regenerative Medicine, 14 Jinhui Road, Shenzhen 518118, China; llilng@sohu.com (C.Z.); zzl@rpxyjy.cn (Z.Z.)

**Keywords:** norcantharidin, immunogenic cell death, autophagy, acidic extracellular pH, bladder cancer

## Abstract

The acidic tumor microenvironment stands as a major obstacle to the efficient elimination of tumor cells. Norcantharidin (NCTD) is a powerful antitumor agent with multiple bioactivities. However, the effect of NCTD under acidic conditions is still unclear. Here, we report that NCTD can efficiently kill bladder cancer (BC) cells in acidic culture, and more intriguingly, NCTD can induce immunogenic cell death (ICD), thereby promoting antitumor immunity. In NCTD-treated BC cells, the surface-exposed calreticulin (ecto-CALR) was significantly increased. Consistently, co-culture with these cells promoted dendritic cell (DC) maturation. The NCTD-induced ICD is autophagy dependent, as autophagy inhibition completely blocked the NCTD-induced ecto-CALR and DC maturation. In addition, the DC showed a distinct maturation phenotype (CD80^high^ CD86^low^) in acidic culture, as compared to that in physiological pH (CD^80^ high CD86^high^). Finally, the NCTD-induced ICD was validated in a mouse model. NCTD treatment significantly increased the tumor-infiltrating T lymphocytes in MB49 bladder cancer mice. Immunizing mice with NCTD-treated MB49 cells significantly increased tumor-free survival as compared to control. These findings demonstrate that NCTD could induce ICD in an acidic environment and suggest the feasibility to combine NCTD with anticancer immunotherapy to treat BC.

## 1. Introduction

Bladder cancer (BC) is among the most prevalent cancers worldwide. In clinical practice, around 80% of BC patients were diagnosed as non-muscle-invasive BC, with good life expectancy [1]. However, despite advancements in research and care, the 5-year survival rate for BC has not markedly changed [1]. Chemotherapy is an important treatment for BC, but it has failed in a large proportion of patients because of the gradual chemoresistance [2]. Therefore, it is necessary to elucidate the mechanism of BC resistance and identify new drugs for BC treatment.

The tumor microenvironment (TME) plays a key in tumor cell survival. Solid tumors are characterized by hypoxia and high glycolytic activity, resulting in increased production and secretion of lactate and H+ to the extracellular space. The elevated glycolysis coupled with poor vascular perfusion leads to an acidic TME, with extracellular pH ranging from 6.5 to 6.9 [3]. The acidic TME creates a physiological barrier for cellular uptake of antitumor drugs, which is termed “ion trapping”, thus leading to drug resistance [4,5]. More importantly, the acidic TME impedes immune cells’ function and contributes to the escape of tumor cells from immune surveillance [6,7]. Therefore, the acidic TME stands as a major obstacle to the efficient elimination of tumor cells. The identification of drugs retaining anticancer activity in an acidic environment may improve the efficacy of cancer therapy.

In addition to direct cytotoxicity, chemotherapeutic agents are particularly efficient when they elicit immunogenic cell death (ICD) [8]. ICD is characterized by the emission of a series of damage-associated molecular patterns (DAMPs), mainly including exposure of calreticulin (CALR) on the cell surface, secretion of adenosine triphosphate (ATP), and release of the chromatin-binding protein high mobility group B1 (HMGB1) [9]. These DAMPs can be recognized by the immune cells and trigger an antitumor immune response [8]. Substantial evidence has indicated that certain chemotherapeutic agents, such as anthracyclines or microtubule-interfering drugs (PTX), can induce ICD [10,11]. However, ICD under an acidic environment is seldom reported.

Norcantharidin (NCTD), a demethylated analog of cantharidin, has been used to treat cancer in China since 1984 [12]. NCTD has powerful and extensive anti-cancer activities towards various types of cancer cells, by inhibiting proliferation, migration and metastasis, and inducing apoptosis [13]. Moreover, NCTD can also improve anti-cancer immunity. NCTD enhanced the activity of tumor-associated macrophages (TAMs) against hepatocellular carcinoma, by inducing a shift from M2 to M1 polarization [14]. Recently, our research has shown that NCTD can increase the tumor infiltration of CD4+/CD8+ T in prostate cancer [15]. Therefore, NCTD has dual anti-cancer activities, namely, direct killing effects and promotion of anti-cancer immunity, implying that NCTD may induce ICD.

In this research, the anticancer activity of NCTD on BC cells was investigated in acidic culture. ICD was tested by CALR exposure and ATP secretion after NCTD treatment. The immunogenic activity of the NCTD-treated cells was valued by stimulation of dendritic cell maturate. Further, the immunogenic activity of the NCTD-treated BC cells was validated in mice. The results show that NCTD efficiently inhibits BC cells in acidic conditions, and it proves for the first time that NCTD could induce ICD of BC cells.

## 2. Results

### 2.1. NCTD Shows Enhanced Cytotoxicity on BC Cells in Acidic Culture

NCTD can inhibit proliferation and induce apoptosis of cancer cells [13]. However, most of these investigations were performed under physiological pH conditions. To mimic the acidic TME, we first tested the proliferation of human BC cell lines (EJ and UMUC3) in acidic medium (pH 6.7). In acidic culture, cells were in a slow-cycling state, as confirmed by decreased DNA replication (Figure 1A), G1 accumulation (Figure 1B), and decreased protein levels of the G1-S regulators, including E2F1, MCM7, and CDC6 (Figure 1C). However, the cells showed no apparent apoptosis (Figure 2B).

NCTD showed enhanced cytotoxicity on BC cells in acidic culture (Figure 2A). The IC^50^ in acidic culture (8.64 μM for EJ, 6.86 μM for UMUC3) was much lower than that in physiological pH medium (22.67 μM for EJ, 32.75 μM for UMUC3). Moreover, NCTD induced obvious apoptosis of BC cells in acidic culture (Figure 2B).

### 2.2. NCTD Increases Surface Exposure of CALR in Both Physiological and Acidic Culture

CALR is a major DAMP that mediates immunogenic cell death. Surface-exposed CALR (ecto-CALR) functions as an “eat me” signal for phagocytosis by dendritic cells [8]. In this study, the effect of NCTD on ecto-CALR was tested by flow cytometry, and PTX and cisplatin (DDP) were used as positive [11] and negative [16] controls, respectively. NCTD and PTX, but not DDP, significantly increased ecto-CALR in both EJ and UMUC3 cells (Figure 3A,B). Acidic culture alone slightly increased the ecto-CALR, but it reached no statistical difference compared with physiological pH (Figure 3A,B). Notably, NCTD induced more apparent ecto-CALR in acidic culture, to a more powerful potential than that in physiological pH (Figure 3A,B). 

Next, the secretion of ATP after NCTD treatment was examined. NCTD did not alter the ATP content in the supernatant and cell lysis, in both physiological and acidic conditions (Figure 3C,D).

### 2.3. NCTD Promotes Autophagy in Both Physiological and Acidic Cultures

Autophagy is involved in the anticancer mechanism of NCTD [13] and participates in the ICD regulation. Therefore, we tested the autophagic profiles after NCTD treatment. NCTD increased the mRNA levels of autophagy-regulating genes, including atg5, atg7, and BECN1 (Figure 4A). Consistently, Western blot analysis showed increased conversion of LC3-I to LC3-II (Figure 4B). The NCTD-induced autophagy was further supported by immunofluorescence which showed obvious RFP-LC3 puncta (Figure 5A,B). 

Next, NCTD-induced autophagy was tested in acidic culture. Acidic culture alone induced RFP-LC3 puncta formation (Figure 5A,B), consistent with previous reports that cancer cells upregulate autophagy as a survival mechanism against acidic stress [17,18]. Intriguingly, in acidic culture, NCTD further increased the formation of RFP-LC3 puncta (Figure 5A,B) and improved the autophagic genes expression (atg5, atg7, and BECN1) (Figure 5C). This indicates that NCTD enhances autophagy in acidic culture.

### 2.4. Inhibition of Autophagy Attenuates NCTD-Induced Ecto-CALR

Autophagy forms the ‘core’ of regulating pathways behind ICD [19]. However, the effect is inconsistent in different reports. On one hand, autophagy may positively regulate ICD by favoring ATP secretion [9,20]. On the other hand, autophagy negatively regulates Hyp-PDT-induced ecto-CALR (without affecting ATP secretion) by attenuating oxidative stress [21]. To elucidate the contribution of autophagy in NCTD-induced ICD, autophagy was inhibited by chemical inhibitors. It shows that chloroquine failed to inhibit autophagy in acidic culture (data not shown). This is consistent with previous research showing that chloroquine lost activity in acidic pH, possibly due to the reduced cellular uptake [18]. While, another autophagy inhibitor, 3-methyladenosine (3-MA), successfully inhibited the NCTD-induced autophagy in acidic culture (Figure 6A,B). Autophagy inhibition by 3MA completely inhibited the NCTD-induced ecto-CALR in acidic culture, as well as in physiological pH (Figure 6C,D). These results indicate that NCTD induces ecto-CALR in an autophagy-dependent manner.

### 2.5. NCTD-Treated BC Cells Promote Dendritic Cell Maturation

To further investigate the immunogenic properties of NCTD-treated BC cells, mouse dendritic cells (mDC) were challenged with MB49 mouse bladder cancer cells pretreated with or without NCTD, then the mDC maturation makers (CD80^+^/CD11c^+^ or CD86^+^/CD11c^+^) were analyzed by flow cytometry. As shown in Figure 7A, MB49 challenge increased the percentage of CD80^+^/CD11c^+^ cell population in both physiological and acidic cultures. As expectation, pretreatment of MB49 with NCTD further enhanced the CD80+/CD11c+ population. While, inhibition of autophagy by 3-MA abolished the NCTD pretreatment-induced CD80^+^/CD11c^+^ enhancement.

The percentage of CD86^+^/CD11c^+^ showed a similar tendency as CD80^+^/CD11c^+^ among groups in physiological pH culture (Figure 7B). However, in acidic culture, there was no significant difference among groups (Figure 7B). This indicates that the acidic pH may suppress MB49 challenge-induced CD86 expression. NCTD alone did not alter the percentages of CD80^+^/CD11c^+^ (Figure 7A) and CD86^+^/CD11c^+^ (Figure 7B), excluding the direct effect of NCTD on DC maturation. Together, these results demonstrate that NCTD-treated MB49 cells can promote DC maturation in both physiological and acidic cultures, in an autophagy-dependent manner.

### 2.6. NCTD Improves Antitumor Immunity in Mice

Finally, we determined the immunomodulatory activity of NCTD in the MB49 mouse bladder cancer model (Figure 8A). NCTD obviously reduced tumor growth in mice (Figure 8C). The proportions of CD4^+^ T and CD8^+^ T cells in peripheral blood were increased after NCTD treatment (Figure 8D,E). Moreover, the immunohistochemistry assay showed increased tumor-infiltrating CD4^+^ T and CD8^+^ T lymphocytes in NCTD-treated mice (Figure 8F,G). 

We further validated the NCTD-induced ICD in mice by using the NCTD-treated MB49 cells as a “tumor vaccine” (Figure 8B). Our results demonstrated that the tumor-free survival among mice immunized with NCTD-treated MB49 cells was significantly increased as compared with the control (immunized with DMSO-treated MB49 cells), indicating that NCTD was a bona fide inducer of ICD in vivo (Figure 8H).

## 3. Discussion

Anticancer chemotherapeutic agents are particularly efficient when they induce ICD thus enhancing antitumor immune response. However, most of the ICD inducers were explored under physiological pH conditions. Since solid tumors are characterized by acidic TME, it is necessary to study the ICD in acidic conditions. In this paper, we show that NCTD is an ICD inducer in both physiological pH and acidic conditions. NCTD can efficiently increase ecto-CALR in BC cells, and co-culture with NCTD-treated BC cells promote DC maturation. Moreover, the mouse in vivo study revealed the enhancing effect of NCTD on antitumor immunity. Further, we show that the NCTD-induced ICD is autophagy dependent. Autophagy underlies the cytotoxicity of NCTD and positively regulates the NCTD-induced ICD. In addition, we reveal that DC shows a distinct maturation phenotype (CD80^high^ CD86^low^) in acidic culture. Our current work not only adds new information on the ICD inducer under acidic conditions but also provides new insights into the anticancer mechanism of NCTD.

### 3.1. NCTD Is an ICD Inducer

It has been long recognized that NCTD has immunomodulatory activity. NCTD can facilitate the activation of macrophages against bacteria in acute peritonitis mouse models [22], decreasing the shift from M2 to M1 polarization of tumor-associated macrophages in hepatocellular carcinoma [14]. Our previous study showed that NCTD can preferentially inhibit the Treg and enhance antitumor immunity of the prostate cancer cell vaccine [15]. In this research, NCTD treatment resulted in increased surface exposure of CALR in BC cells and improved the immunogenicity of MB49 cells. In the mouse bladder cancer model, NCTD treatment increased CD4+ T and CD8+ T frequency in tumor sites and in peripheral blood. Furthermore, immunization with NCTD-treated MB49 cells can protect mice from MB49 rechallenge. Therefore, NCTD clearly represents as an ICD inducer.

Only certain anticancer agents have been reported to induce ICD. ICD inducers have been classified into two classes (type I and type II), on the basis of how they induce endoplasmic reticulum (ER) stress [23]. The majority of ICD inducers are considered type I, which trigger cancer cell death by acting on different targets such as cytosolic proteins (as in the case of shikonin), plasma membrane proteins/channels (as in the case of cardiac glycosides), DNA or DNA replication and repair machinery proteins (as in the case of anthracyclines, mitoxantrone, oxaliplatin, and UVC irradiation); while the ER is affected as a ‘collateral’ effect [23]. On the contrary, type II inducers are considered to lead to cancer cell death by direct targeting ER [23]. So far, only a small number of type II inducers have been recognized, with photodynamic therapy (PDT) as a typical one [10]. NCTD has extensive antitumor effects on diverse cancer cells. However, the specific targets are not determined [13]. NCTD can affect cell growth, apoptosis, autophagy, migration, and invasion through a variety of signaling pathways [13]. There are no reports that NCTD can directly target ER. Therefore, we consider that NCTD should belong to the type I inducer.

In this research, NCTD increases the CALR exposure but does not affect ATP secretion. ICD is characterized by the emission of a series of DAMPs, mainly including CALR, ATP, and HMGB1. Besides, many other DAMPs such as type I IFN, annexin A1, cancer cell-derived nucleic acids, heat shock protein 90 (HSP90), and HSP70 are associated with ICD [8]. In this research, NCTD increases the CALR exposure but does not affect ATP secretion. In the future, the regulatory effect of NCTD on other DAMPs is warranted.

### 3.2. Autophagy Underlies the Cytotoxicity of NCTD and Positively Regulates the NCTD-Induced ICD

Substantial evidence shows that autophagy is involved in the antitumor effect of NCTD. NCTD can induce autophagic cell death in hepatocellular carcinoma [24], human osteosarcoma [25], and neuroblastoma [26]. As known, the acidic tumor microenvironment contributes to drug resistance by creating an “ion trapping” barrier for cellular uptake of antitumor drugs [4,5]. In this study, NCTD shows higher cytotoxicity on BC cells in acidic culture than in physiological pH medium, which indicates that NCTD can enter BC cells in acidic conditions. The enhanced cytotoxicity may arise from the fragile cell conditions in acidic culture. As our results show, in acidic culture, BC cells are in a slow-cycling state with enhanced basal autophagy; while NCTD treatment further enhances autophagy and leads to excessive autophagy. Excessive autophagy has been well documented during ischemia/reperfusion injury, where it can lead to the destruction of essential molecules and organelles, so as to favor cell death [27,28]. Excessive autophagy has also been shown to induce cell death in tumor cells [29]. In this paper, excessive autophagy was associated with enhanced cytotoxicity of NCTD on BC cells in acidic culture. Therefore, induction of autophagy underlies the antitumor activity of NCTD.

In this paper, the NCTD-induced ecto-CALT and the stimulation of NCTD-treated BC cells on DC maturation were blocked by autophagy inhibition, indicating that autophagy positively regulates the NCTD-induced ICD. Actually, reports on the role of autophagy in ICD are not consistent. Chemotherapy-induced autophagy, such as mitoxantrone or thiostrepton, positively regulates ICD by favoring ATP secretion, not altering CALT exposure [9,20]. On the contrary, ROS-induced autophagy negatively regulates CALT exposure by attenuating oxidative stress, leaving ATP secretion unaffected [21]. In our research, the NCTD-induced autophagy was associated with CALT exposure not ATP secretion. Therefore, different autophagy inducers may exert distinct effects on ICD, with autophagy itself not decisive. In addition, the different cell lines used in the study may also account for the discrepancy in the regulation of autophagy on ICD.

Together, our results suggest that induction of autophagy underlies the antitumor activity of NCTD against BC, including the ICD effect. However, although NCTD-induced autophagy is well documented in various cancer cells, the specific mechanism underlying this activity is vague and needs to be further studied [13].

### 3.3. DC Shows Distinct Maturation Phenotype in Acidic Culture

The acidic microenvironment is tightly related to the activation of immune cells. It has been shown that extracellular acidosis increases the expression of HLA-DR, CD40, CD80, CD86, CD83, and CCR7, and triggers the maturation of DC [30,31]. In this study, the percentage of CD80+/CD11c+ and CD86+/CD11c+ was increased in acidic culture. Therefore, the acidic microenvironment could promote DC maturation. However, the MB49 challenge-induced DC maturation showed a distinct phenotype in acidic culture. The expression of CD80 was improved by MB49 challenge in acidic culture. While, CD86 expression showed no significant change before and after MB49 challenge. It seems that the MB49 challenge-induced CD86 expression was inhibited by acidic extracellular pH. Similar results have been reported in other immune cells. The acidic environment downregulated M1-related genes including CD86 and thus promoted macrophage M2 polarization [32]. Another study showed that acidic extracellular pH could suppress CD86 augmentation in THP-1 cells exposed to allergens (2,4-dinitrochlorobenzene and imidazolidinyl urea) [33]. Therefore, the acidic extracellular pH has substantial effects on DC maturation, especially under tumor cell challenge, and should be considered in the development of antitumor immunotherapy strategies.

## 4. Materials and Methods

### 4.1. Cell Culture

The human bladder cancer cell lines EJ, UMUC3, and mouse bladder cancer cell line MB49 were obtained from the ATCC and maintained in Dulbecco’s Modified Eagle’s Medium (DMEM, Gibco, Grand Island, NE, USA) supplemented with 10% fetal bovine serum (Gibco, Grand Island, NY, USA), 100 U/mL penicillin, and 100 U/mL streptomycin. All cells were cultured at 37 °C in a humidified atmosphere of 5% CO_2_. To simulate the acidic tumor microenvironment, the pH value of the cell culture medium was adjusted to 6.6–6.8 by adding HEPES, NaHCO3, and HCl, and determined prior to each experiment to ensure the stability of the culture microenvironment.

### 4.2. EdU Incorporation Assay

To detect cell proliferation ability under an acidic environment, EJ and UMUC3 cells were incubated with 50 μM EdU for 2 h at 37 °C after being cultured in acidic medium for different days, then washed twice with PBS and fixed with 4% paraformaldehyde for 30 min. After incubated with 2 mg/mL glycine for 10 min, the cells were permeated by 0.5% TritonX-100 PBS and stained with Apollo Dyeing reaction solution in the dark for 30 min at room temperature. Permeated again, DNA was stained with 1× Hoechst 33342 reaction diluent in the dark for 30 min at room temperature. The images were obtained by fluorescence microscope.

### 4.3. Cell Cycle Analysis

Cells were collected and washed with PBS, and then fixed with 70% ethanol at −20 °C overnight. After centrifuging at 1000× *g* for 5 min, the cells were re-suspended in PBS for 15 min. Then, the cells were incubated with DNA staining solution containing PI and RNaseA for 30 min at room temperature. Cell cycle distribution was detected by flow cytometry and data were analyzed using FlowJo software (Ashland, OH, USA).

### 4.4. Western Blot

Cells were lysed with RIPA buffer containing PMSF, protease, and phosphatase inhibitors on ice and the protein concentration was determined by the BCA assay. The protein samples were separated by SDS-PAGE and then transferred to PVDF membranes after electrophoresis. After blocking with 5% skim milk in TBST for 2 h, the membranes were incubated with primary antibodies overnight at 4 °C. Finally, the membranes were incubated with HRP-conjugated secondary antibodies for 1 h at room temperature and the target bands were detected by the Chemiluminescence system. The band intensities were quantified using the ImageJ software. The primary antibodies are as follows: LC3 (Cell Signaling, 3868S); P62 (ProteinTech Group, Chicago, IL, USA, 18420-1-AP); β-actin (ProteinTech Group, Rosement, IL 60018, USA, 66009-1-Ig); E2F1 (Cell Signaling, 3742S); CDC6 (ProteinTech Group, Rosement, IL 60018, USA, 11640-1-AP); MCM7 (ProteinTech Group, Rosement, IL 60018, USA, 11225-1-AP).

### 4.5. Cytotoxicity Assay

The cell viability was assessed by MTS assay. EJ and UMUC3 cells were seeded into 96-well plates at a density of 8 × 10^3^ cells/well and cultured in a humidified chamber comprising 5% CO_2_ at 37 °C for 24 h. Then, the cells were treated with NCTD at a serial of 2-fold dilution (0 to 320 μM) in neutral and acidic culture medium. After incubation for 24 h, 10 μL of MTS solution was added to each well, and the cells were incubated at 37 °C for 1 h. Then, the cells were placed in a microplate reader to detect the absorbance at 490 nm, cell viability could be calculated according to the following equation: cell viability = (experimental group OD − zeroing OD)/(control group OD − zeroing OD) × 100%.

### 4.6. Cell Apoptosis Analysis

Cells were collected and washed twice with cold PBS, and then 100 μL of 1× binding buffer was used to resuspend the cells. Next, the cells were incubated with Annexin V-FITC and PI for 15 min in the dark. Then, the cells were washed with cold PBS and resuspended in 200 μL of 1× binding buffer. Finally, the stained cells were analyzed by flow cytometry.

### 4.7. Flow Cytometry

Cells were harvested and washed twice with PBS and then blocked in 1% bovine serum albumin (BSA) for 15 min. Subsequently, cells were stained with fluorescent antibodies for 30 min at 4 °C. PE-labeled anti-Calreticulin antibody (Abcam, Cambridge, UK, ab83220) PE-labeled anti-mouse CD80 antibody (Tonbo Biosciences, San Diego, CA, USA, 50-0801-U025), FITC-labeled anti-mouse CD86 antibody (Tonbo Biosciences, San Diego, CA, USA, 35-0862-U025), and APC-labeled anti-Mouse CD11c antibody (Tonbo Biosciences, San Diego, CA, USA, 20-0114-U025) were used. After washing three times with PBS, the cells were resuspended with PBS and detected by flow cytometry (Becton Dickinson, San Jose, CA, USA) within an hour. Data were analyzed using FlowJo software (Ashland, OH, USA).

### 4.8. Real-Time Polymerase Chain Reaction

Total RNA was extracted from cells using RNAiso plus. After detecting the concentration and purity of total RNA, reverse transcription was performed with the PrimeScript™ RT Master Mix Kit (TaKaRa, Kusatsu, Japan). The reverse transcription procedure was set as 37 °C for 15 min, 84 °C for 5s and 4 °C for a moderate amount of time according to the specification. mRNA expression in cells was measured with TB Green Taq (TaKaRa, Kusatsu, Japan) on the Light Cycler 480 Roche system. Gene-specific primers were listed as follow (5′-3′): β-actin (human): CATGTACGTTGCTATCCAGGC (forward) and CTCCTTAATGTCACGCACGAT (reverse); atg5 (human): AAAGATGTGCTTCGAGATGTGT (forward) and CACTTTGTCAGTTACCAACGTCA (reverse); atg7 (human): CAGTTTGCCCCTTTTAGTAGTGC (forward) and CCAGCCGATACTCGTTCAGC (reverse); BECN1 (human): GGTGTCTCTCGCAGATTCATC (forward) and TCAGTCTTCGGCTGAGGTTCT (reverse); calreticulin (human): CCTGCCGTCTACTTCAAGGAG (forward) and GAACTTGCCGGAACTGAGAAC (reverse). Expression levels of the target genes were normalized against the β-actin gene. In order to ensure the accuracy of the experiment, each experiment needed to repeat three times independently. The result was calculated with the CT value.

### 4.9. ATP Analysis

Extracellular ATP in conditioned medium and intracellular ATP in cell lysate following the indicated treatment were measured by the Enhanced ATP Assay Kit (Beyotime Biotechnology, Shanghai, China), according to the manufacturer’s instruction. Fluorescence was detected by the Spark multimode microplate reader (TECAN, Switzerland). For the intracellular ATP analysis, in order to eliminate the error caused by the difference in cell numbers in sample preparation, the concentration of ATP was converted into the form of nmol/mg protein. Results of each group are presented as the ratio to control.

### 4.10. Immunofluorescence

Cells grown on coverslips were washed three times with phosphate-buffered saline (PBS) very gently and then fixed with 4% paraformaldehyde for 30 min at room temperature. After permeabilizing with 0.2% TritonX-100 for 10 min, cells were blocked in 5% bovine serum albumin (BSA) for 1 h. Subsequently, staining with primary antibodies LC3 diluted 1:200 in 5% BSA was performed overnight at 4 °C and then cells were incubated with the fluorescent secondary antibody of the corresponding species (1:400) for 1 h at room temperature. After the final wash (three times for 5 min) in PBS, cells were counterstained with DAPI for 5 min. The coverslips were mounted on glass slides with mounting medium and Antifade Solution. Fluorescence images were acquired using a confocal fluorescence microscope (Zeiss LSM-880) and images were processed with ZEN software.

### 4.11. BMDC Culture and Stimulation

Bone marrow-derived DCs (BMDCs) were generated from the femurs and tibias of 5–7-week-old C57BL/6 mice and the bone marrow cells were harvested and treated with ACK lysis buffer for 5 min. Then, the cells were washed twice with PBS and resuspended with complete medium. In all, 2 × 10^6^ bone marrow cells were cultured with 10 mL of complete medium supplemented with 20 ng/mL GM-CSF for 3 days. Then, an additional 10 mL of complete medium was added and supplemented with 20 ng/mL GM-CSF. On day 6, half of the medium was replaced with fresh complete medium, and the cells were maintained. On day 8, immature DC were acquired and co-cultured with MB49, NCTD-pretreated MB49, or NCTD+3MA-pretreated MB49 for 24 h. CD80^+^/CD11c^+^ or CD86^+^/CD11c^+^DCs populations were detected by flow cytometry.

### 4.12. Animal Experiments

All mice were maintained in pathogen-free conditions and the experiments were performed according to the guidelines of the Animal Experimentation Ethics Committee at Southern Medical University (Guangzhou, China). The ethical code is SMUL2016022. To establish the tumor model, C57BL/6/female mice (6–8 weeks, 20 ± 2 g) were injected subcutaneously with 5.0 × 10^6^ MB49 cells in 0.1 mL of PBS. When palpable tumors were present, the mice were randomly divided into control and NCTD treatment groups. The mice in the control group were given a 0.1 mL dose of PBS solution by intraperitoneal injection, those in the treatment group were administered with NCTD at a dose of 10 mg/kg/day for 14 days. Tumor volume was measured every 3 days. On the 30th day post-inoculation, the mice were sacrificed, and the tumors were obtained to detect relevant indicators. The blood samples of each mouse were collected and mixed with ACK lysing buffer (Solarbio) to lyse red blood cells. After washing with PBS, the cells were stained with APC-Cy7-A-labeled anti-mCD4 or FITC-labeled anti-mCD8 antibody for 30 min. Then, cells were washed three times with PBS, resuspended with PBS, and detected by flow cytometry (Becton Dickinson) within an hour. Data were analyzed using FlowJo software (Ashland, OH, USA).

For tumor vaccination assay, MB49 cells were incubated with NCTD for 48 h, and those treated with DMSO were used as the control group. Next, cells were fixed in 30% ethanol (*v*/*v*) for 1 h at room temperature. Cell viability was tested by trypan blue (0.4%) staining. Then, the cells (5 × 10^5^ cells per mouse) were injected subcutaneously to the left flank of immunocompetent C57BL/6 mice. After 7 days, the vaccinated mice were re-challenged with untreated MB49 cells (5 × 10^6^ cells/mouse) by intraperitoneal injection. The ascitic fluid development was observed to monitor tumor growth. After 60 days, all mice were sacrificed to examine the peritoneal tumor development.

### 4.13. Immunohistochemical Analysis

Tumor tissues of mice were collected and fixed with 4% paraformaldehyde, and then embedded in paraffin. Paraffin sections (5 μm) were deparaffinized, rehydrated with PBS, and incubated with 3% H_2_O_2_ for 10 min. Then, antigen retrieval was achieved by covering the slides with citrate buffer (pH 6.0) and heating for 10 min at 95 °C. The sections were blocked with 5% BSA for 30 min at room temperature and incubated with anti-mCD4 (1:700; Abcam, Cambridge, UK), and anti-mCD8 (1:1000; eBioscience, San Diego, CA, USA) antibody at 4 °C overnight. After washing with PBS, the sections were incubated with the secondary antibody conjugated to diaminobenzidine (DAB) for 1 h at room temperature. All tissues were counterstained with hematoxylin.

### 4.14. Statistical Analysis

Statistical analysis software SPSS 22.0 was used for data analysis and data are expressed as the mean ± standard deviation (SD). Statistical comparisons between two groups were performed utilizing a two-tailed Student’s *t*-test and one-way analysis of variance (ANOVA) for multiple groups. *p*-value < 0.05 was considered statistically significant.

## Figures and Tables

**Figure 1 ijms-23-03944-f001:**
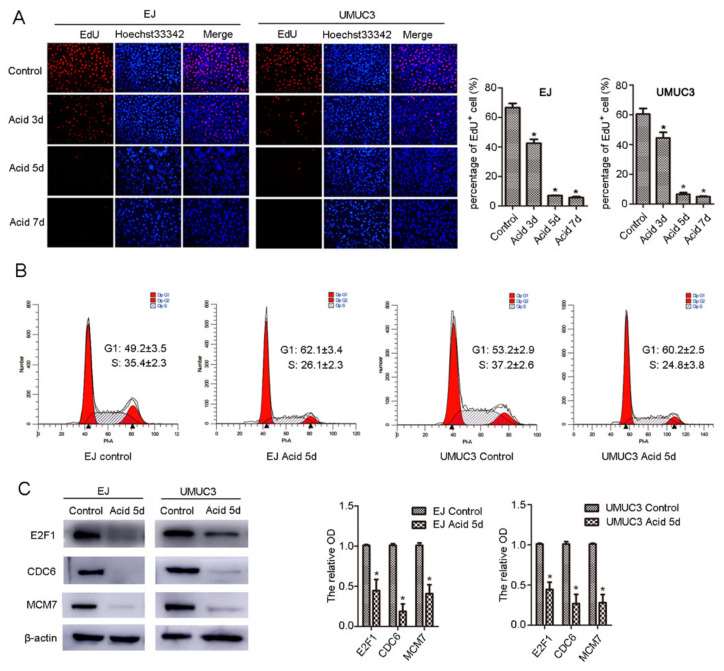
Bladder cancer cells enter into low-cycling state in acidic culture. (**A**) EJ and UMUC3 cells were cultured in acid medium for 3, 5, and 7 days, DNA replication was detected by EdU incorporation and observed under fluorescence microscope (100×). (**B**) Cells were cultured in an acid medium for 5 days, cell cycle distribution was detected by flow cytometry, and (**C**) E2F1, CDC6, and MCM7 protein levels were evaluated by western blot. β-actin was used as an internal control. Data are presented as the mean ± SD (*n* = 3), * *p* < 0.05 vs. the control group.

**Figure 2 ijms-23-03944-f002:**
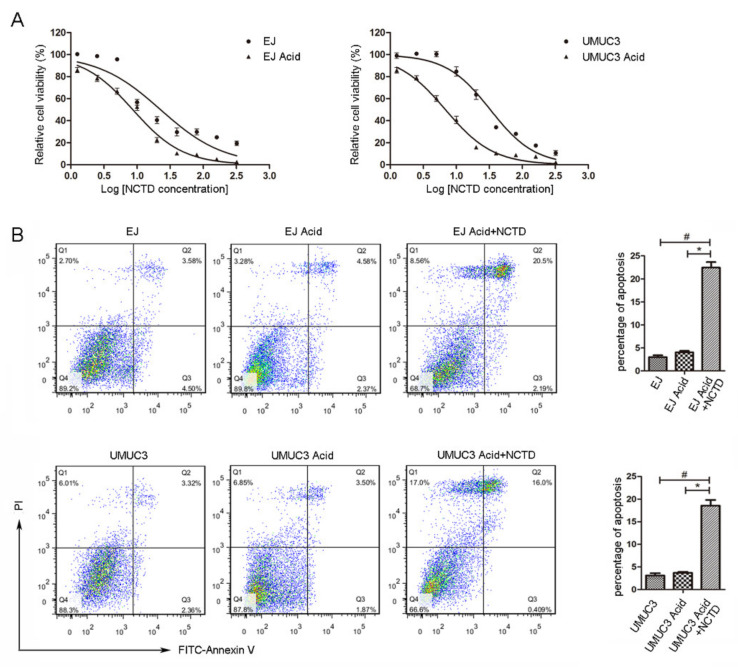
NCTD induces apoptosis of bladder cancer cells under acidic environment. (**A**) EJ and UMUC3 cells were treated with an increasing concentration of NCTD (0–320 μM) for 24 h in physiological (pH 7.2–7.3) and acidic culture medium (pH 6.6–6.8). Cell viability was measured by the MTS assay. The logarithm of the NCTD concentration was taken, and then a curve was fitted to calculate the IC50 value. (**B**) EJ and UMUC3 cells were treated with NCTD (3 μM) in acidic medium, cell apoptosis was measured with Annexin-V/PI double-stained flow cytometry. Data are presented as the mean ± SD (*n* = 3), # *p* < 0.05 vs. the control group, * *p* < 0.05 vs. the acid group.

**Figure 3 ijms-23-03944-f003:**
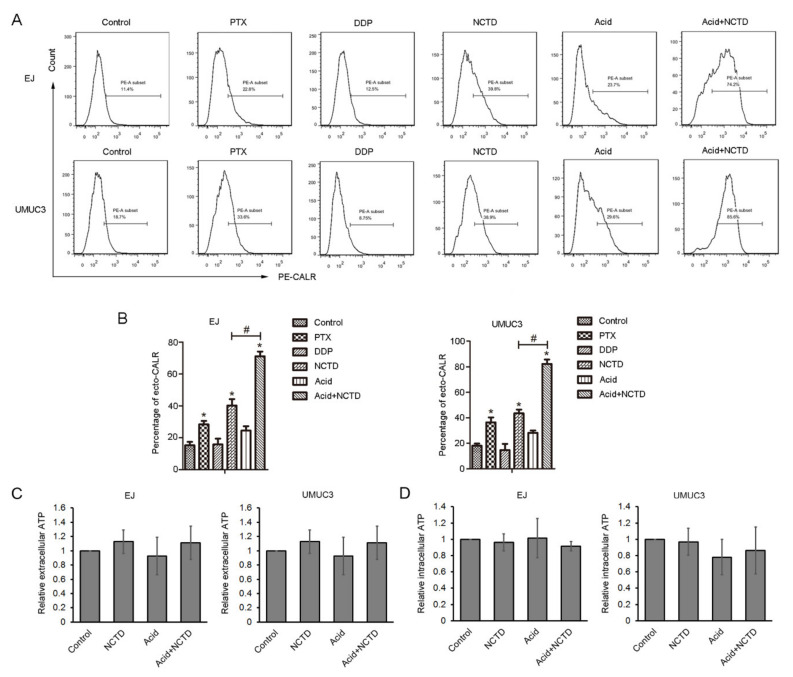
NCTD promotes calreticulin (CALR) surface exposure in bladder cancer cells. (**A**) Flow cytometry analysis of CALR on the cell surface of EJ and UMUC3 bladder cancer cells treated with NCTD for 3 days in physiological (12 μM NCTD) and acidic culture medium (3 μM NCTD). Paclitaxel (PTX) and Cisplatin (DDP) were used as positive and negative controls, respectively. (**B**) Quantification of the percentage of ecto-CALR. Data are presented as the mean ± SD (*n* = 3), * *p* < 0.05 vs. the control group, # *p* < 0.05 vs. the NCTD group. Extracellular (**C**) and intracellular ATP levels (**D**) were analyzed, data are presented as relative ATP abundance to the control cells.

**Figure 4 ijms-23-03944-f004:**
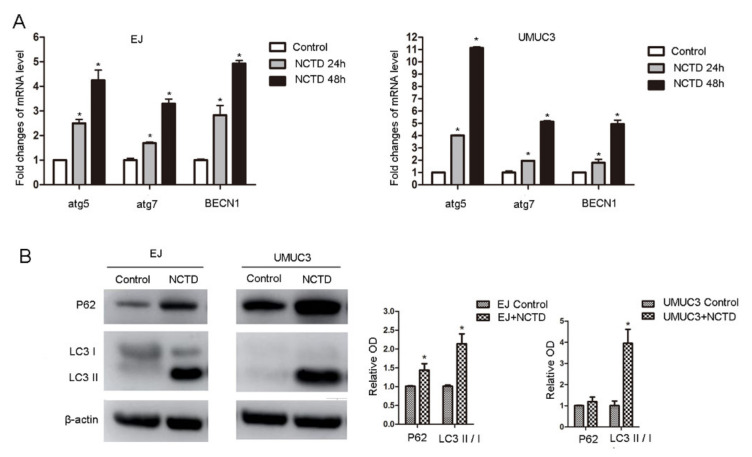
NCTD promotes bladder cancer cell autophagy. (**A**) EJ and UMUC3 cells were treated with NCTD (12 μM) for 24 h and 48 h, mRNA levels of atg5, atg7, and BECN1 were detected by real-time PCR. (**B**) EJ and UMUC3 cells were treated with NCTD (12 μM) for 48 h, P62 and LC3 protein levels were evaluated by Western blotting, β-actin was used as an internal control and relative protein levels were measured. Data are presented as mean ± SD (*n* = 3), * *p* < 0.05 vs. the control group.

**Figure 5 ijms-23-03944-f005:**
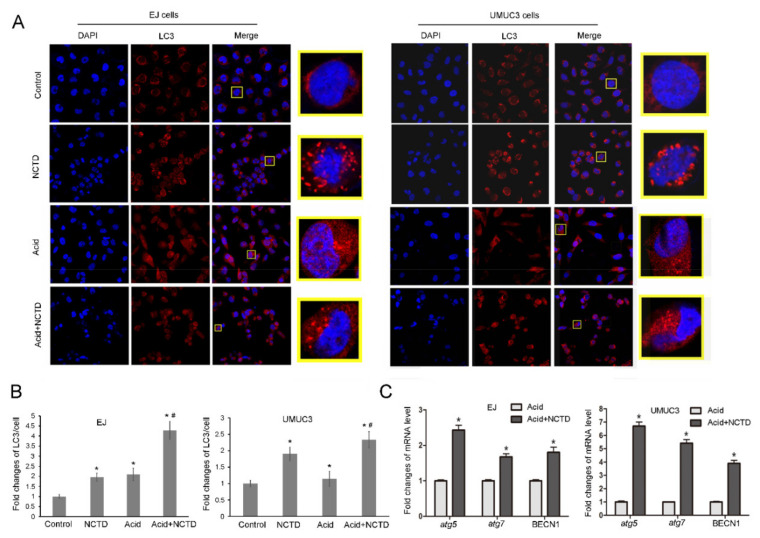
NCTD further enhances autophagy in acidic culture. EJ and UMUC3 cells were treated with NCTD under physiological pH or in an acidic medium for 24 h, (**A**) autophagosomes were detected by using immunostaining with LC3 antibody and observed under fluorescence microscope (400×). (**B**) the relative LC3 puncta/cells were presented as fold changes. * *p* < 0.05 vs. the control group; # *p* < 0.05 vs. the acidic group. (**C**) mRNA levels of atg5, atg7, and BECN1 were detected by Real-time PCR. Data are presented as the mean ± SD (*n* = 3), * *p* < 0.05 vs. the Acid group.

**Figure 6 ijms-23-03944-f006:**
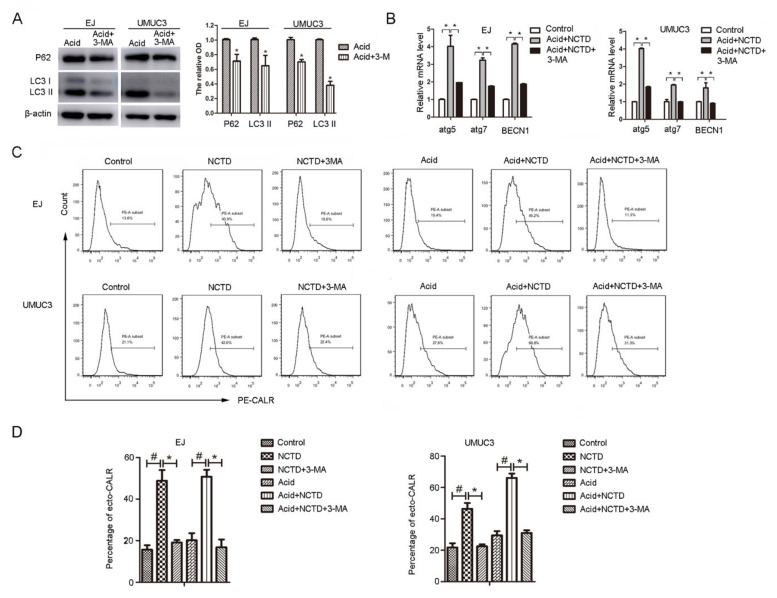
Inhibition of autophagy decreased the ecto-CALR induced by NCTD. (**A**) EJ and UMUC3 Cells were exposed to 3MA (5 mM) in acidic medium, P62 and LC3 protein levels were evaluated by Western blotting, β-actin was used as an internal control and relative protein levels were measured. (**B**) Cells were treated with NCTD (3 μM) with or without 3-MA (5 mM) in acidic medium for 24 h, mRNA levels of atg5, atg7, and BECN1 were detected by Real-time PCR. Data are presented as the mean ± SD (*n* = 3), * *p* < 0.05 vs. the acid group. (**C**) Flow cytometry analysis of ecto-CALR exposure on the cells treated with NCTD with or without 3-MA (5 mM) in physiological and acidic culture. (**D**) Quantification of the percentage of ecto-CALR. Data are presented as the mean ± SD (*n* = 3), * *p* < 0.05 vs. the NCTD group, # *p* < 0.05 vs. the control group.

**Figure 7 ijms-23-03944-f007:**
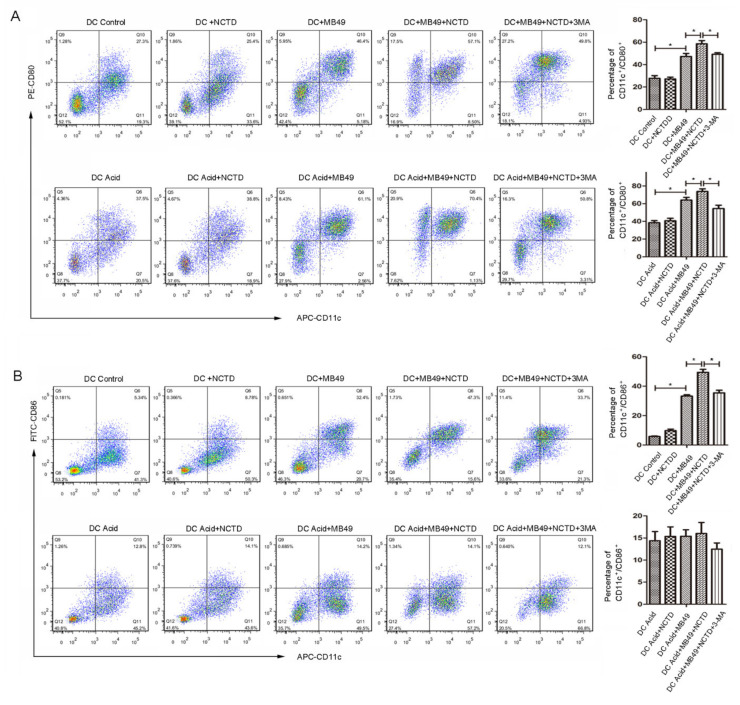
NCTD-treated bladder cancer cells promote DC maturation. MB49 cells were incubated with or without NCTD in physiological (12 μM NCTD) and acidic culture medium (3 μM NCTD) for 24 h, followed by co-culture with BMDCs for 24 h. BMDCs treated by NCTD alone were served as control group. The cells were then labeled with CD11c, CD80 and CD86, CD11c^+^/CD80^+^ (**A**), and CD11c^+^/CD86^+^ (**B**) double-positive cells were measured using flow cytometry. Quantification of the percentage of double-positive cells. Data are presented as the mean ± SD (*n* = 3), * *p* < 0.05.

**Figure 8 ijms-23-03944-f008:**
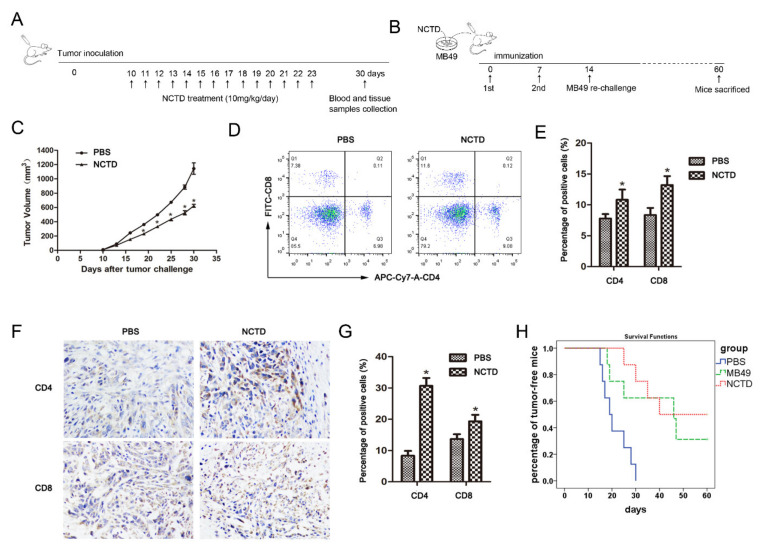
NCTD improves antitumor immunity in mice. (**A**) The in vivo immunomodulatory activity of NCTD was tested by MB49 mouse bladder cancer model. (**B**) The in vivo tumor immunization assay was performed by using NCTD-treated MB49 cells as a “tumor vaccine”. (**C**) Tumor volumes of MB49 cell xenograft mice in (**A**) model. (**D**,**E**) Blood samples were collected from mice and stained with APC-Cy7-A-labeled anti-mCD4 or FITC-labeled anti-mCD8 antibody. The percentage of CD4^+^ T or CD8+ T in the (**A**) model was analyzed by flow cytometry. (**F**,**G**) Immunohistochemistry and quantitative analyses of CD4^+^ T and CD8^+^ T in the tumor tissues in (**A**) model. Data are presented as the mean ± SD (*n* = 3), * *p* < 0.05 vs. the control group. (**H**) Tumor-free survival analysis by Kaplan–Meier curves of mice immunized with NCTD-treated MB49 cells in (**B**) model. The experiment was repeated twice.

## Data Availability

Not applicable.

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
