# Peer review of "Norcantharidin Induces Immunogenic Cell Death of Bladder Cancer Cells through Promoting Autophagy in Acidic Culture"

_ijms, 2022, doi:10.3390/ijms23073944_

Round 1

Reviewer 1 Report

The manuscript is well written, and the study performed is novel. Many published articles on NCTD show work on bladder cancer, but no one reported the effect of pH (acidic) induces ICD in bladder cancer. I like this work very much and approve of the acceptance. However, the author needs to address a few points which must be known to the readers.

  1. Are the cell lines used isolated from the patients or procured from the company?
  2.  As we know, the tumor microenvironment creates acidic pH, and hence for efficient treatment, the drug/compound should be attracted towards that site (tumor). What are the plausible mechanisms by which NCTD targets more on tumors while they cultures in acidic environments?
  3. Figure 4A, 5B Y-axis label should be changed to 'Fold Changes.'
  4. There is no magnification mentioned in any figure legends. Figure 4C and 5A can be merged.
  5. The quality of all figures, especially microscopy images, is inferior. Few pictures are captured at different magnifications (i.e., EJ control vs. EJ + NCTD).
  6. The author needs to make an additional figure in Figure 8, in which a schematic diagram must be included. Presently, in vivo study looks very confusing.

Reviewer 2 Report

The manuscript focuses on the induction of immunogenic cell death by noncantharidin in acidic culture conditions of bladder cancer cells. through promoting autophagy. The study is interesting, well-organised and well-performed. However, some issues should be addressed:

  1. Did the authors detect release of HMGB1? Since the ICD criteria are CALR exposure, ATP secretion and HMGB1 release (as mentioned in line 47-50), the authors should discuss these criteria in relation to their results in the Discussion section in 3.1.
  2. In the legend of Figure 8 and in 2.6 section please provide information about the time point of CD4 and CD8 T cell detection/immunohistochemistry analyses.
  3. In Material and Methods please provide a separate section on Ethics (regarding the in vivo experiments).
  4. I would suggest to add a paragraph in the Discussion section (before 3.1) summarising the main results.
